# Heavy Metals in Groundwater of Southern Italy: Occurrence and Potential Adverse Effects on the Environment and Human Health

**DOI:** 10.3390/ijerph20031693

**Published:** 2023-01-17

**Authors:** Maria Triassi, Pellegrino Cerino, Paolo Montuori, Antonio Pizzolante, Ugo Trama, Federico Nicodemo, Jacopo Luigi D’Auria, Sabato De Vita, Elvira De Rosa, Antonio Limone

**Affiliations:** 1Department of Public Health, “Federico II” University, Via Sergio Pansini nº 5, 80131 Naples, Italy; 2Istituto Zooprofilattico Sperimentale del Mezzogiorno, Via Salute nº 2, 80055 Portici, Italy; 3General Directorate of Health, Campania Region, Centro Direzionale is. C3, 80143 Naples, Italy

**Keywords:** groundwater, heavy metals, risk assessment, contaminant loads, Jenks methods, principal component analysis

## Abstract

This study reports the data on the contamination caused by heavy metals in the groundwater of the Campania Plain (CP) in Southern Italy. A total of 1093 groundwater samples were obtained from the following aquifers: coastal plains (GAR, VCP, VES, SAR, and SEL), volcanic districts (PHLE and VES), and carbonate massifs (MAS and LAT). In this study, the investigation depth ranged from 5 m (GAR) to 200 m (PHLE). The sequence of heavy metal content in groundwater samples was B > Fe > Al > Mn > Zn > Ba > Ni > As > Cu > V > Se > Pb > Cd. The heavy metal pollution index (HPI) and heavy metal evaluation (HEI) demonstrated that the study areas in which groundwater samples were sampled are not risk zones. Moreover, health risk assessment shows that hazard index (HI) values for heavy metals were found to be significantly low in groundwater samples. In non-carcinogenic risk evaluation for the adult group, the risk was low, whereas for children and infants, the risk was >1 for arsenic alone. Carcinogenic risk assessment (CR) was found lower for adults, children, and infants. The Jenks optimization method was used to evaluate the distribution of heavy metals in the groundwater of CP, and the principal component analysis technique (PCA) was employed to determine the source of heavy metals, and it was found that mixed sources (natural and anthropogenic) may be responsible for heavy metals presence.

## 1. Introduction

Groundwater (GW) is a very important natural resource for human water needs, especially given the good quality of groundwater [1]. The good quality of GW is due to the natural protection of the soil, which acts as a mechanical, physical, and chemical filter; however, in areas where population density is high and human activity is intensive, GW is mainly exposed to pollution [2,3]. Moreover, compared to surface water (SW), natural filling of GW is an extremely slow process. For this reason, GW must be monitored and protected with increasing attention [4].

Heavy metals exist naturally in the ecosystem, and their existence in groundwater is typically undesirable because many of them have toxic effects, even at low amounts. This is particularly difficult in metropolitan and agricultural areas where groundwater acts as the main source of water reserves. Arsenic accumulation in groundwater is a vastly known universal problem affecting a large number of people in numerous countries. Lead, mercury, and cadmium in groundwater have also negatively impacted public health and the environment. The World Health Organisation (WHO) has included these four heavy metals (arsenic, lead, mercury, and cadmium) in its list of the top ten chemicals of special concern to the public because of their high toxicity, longevity in the ecosystem, and bioaccumulative properties [5]. In recent years, environmental pollution by heavy metals (HM) has been a growing public health and environmental concern. Even though the earth’s crust contains heavy metals, manmade or natural sources may be to blame for groundwater pollution. The usual factors that control the presence of HM in groundwater are the chemical–physical features of the aquifer, precipitation periodicity, the quality of infiltrating water, and the duration of residence [6,7,8,9,10]. Wastes from various industrial processes (such as tanning, electroplating, manufacturing of chemicals and textiles, mining, smelting, etc.), soil pollution, underground storage tanks, landfills, tailings ponds, municipal wastewater, contaminated surface water, agricultural fertilizers and pesticides, etc., are all attributed to anthropogenic origins [11,12]. Some HM are indispensable for the physiological and biochemical functionality of flora, fauna, and humans, whereas other HM have toxic effects even in small amounts. Therefore, from the public health point of view, these HM are divided into essential (iron, manganese, zinc, copper, etc.) and non-essential elements (lead, cadmium, arsenic, etc.). Three main channels—inhalation, ingestion, and cutaneous ingestion—allow large concentrations of HM to interact with people. Of these human exposure pathways, it is important to manage ingestion through drinking water and food preparation as well as skin contact through household activities with contaminated water.

Principal components analysis (PCA), among other statistical techniques, has been used to identify the sources of metals and their connections to groundwater contamination [13,14,15]. In this paper, the amounts of the principal heavy metals are studied to detect changes due to human pollution in the geochemical background of the Campanian Plain (CP) in Southern Italy. The CP is a good case study area because its geology and hydrology are well understood, and it exhibits significant spatial, hydrogeological, and geochemical heterogeneities [16,17]. This region has recently gained a reputation for being used by local criminal organizations to illegally dispose of poisonous and radioactive garbage. Due of the unlawful burning of waste in the area after these findings, the region has been given the unsettling name “Terra dei Fuochi” (Land of Fires) [18,19,20].

This study is part of a large project aimed at contributing to the knowledge of the pollution affecting the GW of the Campania Plain, Southern Italy. The objective of this project is to assess the GW pollution due to effluents from local industries, agriculture, and the urban impact by identifying several groups of organic and inorganic chemicals and some indicators of microbial pollution. To the best of our knowledge, no previous studies report the contamination of GW by HM in Southern Italy and in the Campania Plain, but some geostatistical studies were previously carried out in the study area to map groundwater contamination risk. In fact, this region is among the most studied in Southern Italy [21]. Based on the aforementioned factors, the main aim of this paper was to (1) characterize GW contamination by heavy metals in the Campania Plain; (2) analyze factors controlling spatial patterns of heavy metals; (3) assess the environmental risk of GW pollution by heavy metals; (4) evaluate the potential adverse effects of heavy metals on human health; and (5) provide a guide in policy formulation toward the restoration of the Campania Plain and to create a starting point about a future study on the GW pollution by heavy metals in this area.

## 2. Materials and Methods

### 2.1. Study Area: Campania Plain, Southern Italy

The Campanian Plain (CP) is a ~1400 km^2^ region with more than 1 million inhabitants. It is bounded by Monte Massico and the Roccamonfina volcano to the north, the carbonate massifs of Maggiore, Tifatini, Durazzano, and Avella to the east, the Campi Flegrei and Vesuvius volcanic systems to the south, and a ~50 km long coastal plain along the Tyrrhenian coast on the western boundary [17,22]. A detailed map of CP geology is shown in Figure 1. The area exhibits heterogeneous land use: (i) the urban area makes up about 20% of the total area; (ii) the agricultural land makes up about 75% of the total area (where a variety of different crops are grown); and (iii) the remaining area is made up of forests and pastures, mostly on the ridges of the massifs that surround the plain. High geographical, hydrogeological, and geochemical heterogeneities define the CP. In fact, a number of processes related to the presence of several geological units and human activity have an impact on the mineralization of groundwater, the main source of water within the CP.

The CP is a particularly vast graben that was created during the Pliocene and collapsed during the Quaternary. It was then filled by volcanic products from Mount Roccamonfina, the Phlaegrean Fields, and Mount Vesuvius, as well as by marine and alluvial deposits produced by the weathering and decay of both the pyroclastic deposits mantling and the carbonatic rocks forming the surrounding mountains [23,24,25]. The marine clays and sands of the overlying unit are made of Phlegrean volcano-clastites and volcanic sediments from other volcanic centers. These marine clays and sands are made of trachytic–phonolithic pyroclastic materials, sand, and cinerite reworked in a subaqueous environment, and they cover the carbonate bedrock of the Apennine chain with an upper limit between 90 and 20 m below sea level [26]. On the basis of its hydrogeological features, the study area can be divided into two interconnected groundwater bodies related to the Volturno River plain and the Phlegrean Fields pyroclastic hills [27].

Due to activity in the Phlegrean Fields, the outcropping unit in the southern portion of the plain is an uneven succession of pozzolans, cinerites, and sandy pyroclastites; at the boundaries of the plain, the carbonate aquifers are the most important [17,28]. The shallow aquifer in the study region is primarily housed in volcanic and alluvial sediments; it receives lateral inflows from karst and volcanic aquifers at the eastern and northern margins of the CP that replenish it. Groundwater in the region moves from east to west. Numerous hydrostratigraphic systems, including quaternary alluvial deposits, pyroclastic deposits, carbonate karstified systems, and silico-clastic systems, are identified by the Campania Plain’s complex lithology and structural makeup [29]. The Mesozoic limestone and subordinately dolomite aquifers, which constitute the most notable mountain ranges and the main sources of drinking water in the area, are among the major aquifers. The Somma–Vesuvius and Phlegrean Field volcanoes’ ash-fall pyroclastic soils, which govern the growth of the epikarst zone, are another source of hydrogeological oddity [30]. The second groundwater resource in the area is represented by alluvial aquifers, which are often medium to highly permeable and form regional aquifer systems. These aquifers can be divided into internal and costal alluvial types.

Other significant aquifers and groundwater resources in the area are represented by the volcanic formations of Ischia Island, Roccamonfina, Somma–Vesuvius, Phlaegrean Fields, and Roccamonfina. Because of the priceless thermal and mineral fluids they contain, these aquifers are significant economic assets. Terrigenous Miocene–Pliocene turbidite, molasse, and clastic series, as well as Cretaceous–Paleogene basin series, which outcrop primarily in the interior areas, are examples of a minor type of aquifers [31].

For this study, water samples were sampled in the following aquifers with respect to their sampling environment into coastal plains (GAR, VCP, VES, SAR, and SEL), volcanic districts (PHLE and VES) and carbonate massifs (MAS and LAT). Due to the large extension and complex hydrogeological settings of the Campania region, a regional map of piezometric levels and the map of the inherent depth of the water table were not available. However, in this study, the investigation depth ranged from 5 m (GAR) to 200 m (PHLE).

### 2.2. Sampling

For the evaluation of heavy-metal contamination in the groundwater of the Campania plain, several samplings were executed during the year 2015. To these aims, more than 1000 samples of groundwater were obtained from different wells in the study area in pre-washed polyethylene bottles. Dynamic sampling with preliminary purge was performed. For heavy metal analysis, water samples were filtered using Whatman filter papers and immediately acidified with concentrated HNO_3_. More than 20 heavy metals were investigated using a Thermo Scientific inductively coupled plasma mass spectrometer (ICP-MS) (Thermo Fisher Scientific, Braunschweig, Germany). In situ, physical parameters such as pH, temperature, and electrical conductivity (EC) were measured using XS PC 70 Vio sensors (XS, Switzerland) (Appendix A). Samples were located in an icebox and carefully transported to the laboratory and stored in a deep freezer (−4 °C) for further analysis.

### 2.3. Sample Cleanup and Analysis, Instrumental Analysis and Quality Assurance and Control

The analyses were conducted using a Thermo Scientific^TM^ ICAP^TM^ RQ inductively coupled plasma mass spectrometer (Q-ICP-MS), operated by Qtegra^TM^ Software (Version 2.7.2425.65), with a Burgener Mira-Mist nebulizer, a Quartz cyclonic spray chamber refrigerated to 2.7 °C, and skimmer cones. The working conditions of the Q-ICP-MS equipment were developed using a tuning solution (Ba, Bi, Ce, Co, In, Li, U 1.00 µg/L, Thermo Scientific) on masses ^115^In, ^7^Li, ^59^Co, ^238^U, ^209^Bi, and ^140^Ce were utilized for oxide and doubly charged interference checks. The analysis was conducted in KED (Kinetic Energy Discrimination) mode using helium as the collision gas, and the parameters were plasma gas flow (Ar): 14.8 mL/min; nebulizer gas flow: 0.98 L/min; auxiliary gas flow: 0.85 L/min; ICP RF Power: 1550 W; and CeO/Ce = 0.0057. Cell gas flow was 4.8 mL/min for He.

The Q-ICP-MS was used to estimate the concentrations of Al, Sb, Ag, As, Be, Cd, Co, Cr, Fe, Hg, Ni, Pb, Cu, Se, Mn, Ti, Zn, B, Ba, Mo, Sn, and V in groundwater samples. Each sample was evaluated in triplicate, and each was examined in triplicate by Q-ICP-MS detection [32,33].

The calibration standards were prepared with multielement standard solution CertiPUR^®^ (Merck, Darmstadt, Germany) 1000 mg L^−1^ at concentrations: 0.5, 1.0, 2.5, 5.0, 10.0 μg/L^−1^ for Hg and 0.5, 1.0, 2.5, 5.0, 10.0, 50.0, 200.0, and 500.0 μg/L^−1^ for V, Mn, Fe, Ni, Cu, Zn, B, Al, As, Se, Sb, Cd, and Ba. An internal standard mix comprising 50 μg/L^−1^ Ge, 5 μg/L^−1^ Ir, 10 μg/L^−1^ In, and 25 μg/L^−1^ Y was introduced online with an internal standard mixing kit. The internal standard elements were appropriately matched to analyte elements [2,34].

### 2.4. Mapping Technique of Groundwater

The mapping technique of groundwater used in this study is the Jenks approach, which uses natural breaks. Natural groupings included in the data serve as the foundation for natural break classes. There are identified class breaks. It is advisable to group values that have a substantial amount in common and to emphasize class distinctions. The features are separated into classes whose borders are determined by the degree of variation in the data values. This technique aims to find the optimal grouping of values into several groups. This is achieved by aiming to maximize each class’s divergence from the means of the other groups while minimizing the average deviation of each class from the class mean. In other words, the approach aims to increase the variation between classes while decreasing the variance between classes.

### 2.5. Evaluation Indices

Understanding the extent of local heavy metal contamination is crucial for the responsible and safe use of groundwater resources. To gauge the degree of water contamination, heavy metal pollution (HPI), and heavy metal evaluation (HEI), indices are frequently utilized [35,36].

#### 2.5.1. Heavy Metal Pollution Index (HPI)

The heavy metal pollution index, which was developed by Mohan et al. (1996), represents the composite influence of heavy metals on groundwater quality. This index establishes a rating or weightage (*W_i_*) between 0 and 1 and is inversely proportional to the standard permissible value (*S_i_*) of the corresponding parameter [37,38,39].

The HPI evaluation model is expressed by Equations (1) and (2) as follows:(1)$$\mathrm{HPI}=\frac{\sum_{i=1}^{n}W_{i}Q_{i}}{\sum_{i=1}^{n}W_{i}}$$
where *W_i_* is denoted as the unit weight of the *i*th parameter, *Q_i_* is the sub-index of the *I*th parameter, and *n* is the number of heavy metals measured. *Q_i_* is expressed as follows:(2)$$Q_{i}=\sum_{i=1}^{n}\frac{|M_{i}-I_{i}|}{S_{i}-I_{i}}\times 100$$

If *S_i_* is the highest allowable concentration for the *i*th parameter based on the water quality standard, *I_i_* is the ideal value or permissible limit for the *i*th parameter, and *M_i_* is the measured metal value for the *i*th sample. *W_i_* is set to 1/*S_i_* [36]. According to HPI results, the pollution category was constituted of three grades as follows: low (HPI < 50), medium (HPI ¼ 50 e100), and high (HPI > 100) contamination [35].

#### 2.5.2. Heavy Metal Evaluation (HEI)

Like HPI, HEI is also another method of assessing the overall water quality [40]. HEI is calculated by the following equation:(3)$$\mathrm{HEI}=\sum_{i=1}^{n}=\frac{C_{i}}{S_{i}}$$
where *S_i_* is the maximum permissible concentration for the *i*th parameter, and *C_i_* is the *i*th parameter’s measured value. The HEI evaluation value is also used to grade the pollution level, and it is obtained as follows: pollution levels can be classified as low (HEI 40), medium (HEI ¼ 40 e 80), and high (HEI > 80) [41].

### 2.6. Human Health Risk Assessment

Health risk assessment is executed to evaluate the health risk of a person due to exposure to a factor by evaluating the possibility of negative effects on the human body [42]. In accordance with the International Agency for Research on Cancer (IARC), the heavy metals of Cr, Cd, and As ions are classified as carcinogenic pollutants, whereas Al, Ba, Mn, Fe, Ni, Cu, Zn, Ba, and Pb compounds are classified as non-carcinogenic [43]. In this study, the carcinogenic risk assessments were applied to define Cd, Cr, and As ion contents in samples, whereas non-carcinogenic risk assessments were used to study other metals.

#### 2.6.1. Non-Carcinogenic Health Risk Assessment

Non-carcinogenic health risk evaluations are commonly used to assess chronic daily intake (*CDI*) and hazard quotient (HQ) as follows:
(4)*CDI* = (C × IR × EF × ED)/(BW × AT)
where *CDI* indicates the average dose contacted through ingestion, C ¼ heavy metal concentration (mg/L), IR ¼ ingestion rate (2 L of water/day for adults and 1 and 0.75 L/day for children and infants, respectively), ED ¼ exposure duration (30 years), EF ¼ exposure frequency (365 days/year), and AT ¼ average exposure time (ED × 365). BW refers to the average body weight in kg: 60, 10, and 5 kg for an adult, child, and infant, respectively (U.S. Environmental Protection Agency, USEPA, 2005) [44].

The HQ for non-carcinogenic risk can be calculated by the following equation [45]:(5)$$\mathrm{HQ}=\frac{CDI}{RfD}$$
where, based on the USEPA database, the oral toxicity reference dose values (*RfD*) are 0.0003 mg/kg-day for As, 0.14 mg/kg-day for Al, 0.0005 mg/kg-day for Cd, 0.003 mg/kg-day for Cr, 0.04 mg/kg-day for Cu, 0.02 mg/kg-day for Ni, 0.0014 mg/kg-day for Pb, 0.3 mg/kg-day for Zn and Fe, 0.005 mg/kg-day for Mo, 0.14 mg/kg-day for Mn, 0.0003 mg/kg-day for Co, and 0.2 mg/kg-day for Ba, respectively [44]. Chronic risk level scale (HQ) based on typical daily intake (*CDI*) and reference dose (mg/kg-day) is categorized based on the ratio of *CDI*/*RfD* indicating if ≤1 (no risk), if >1 ≤5 (low risk), if >5 ≤10 (medium risk), and if >10 (high risk) [42].

#### 2.6.2. Carcinogenic Health Risk Assessment

Human cancer risk is increased when heavy metals are consumed in water over an extended period of time. Therefore, it is crucial to evaluate the health hazards associated with carcinogens. The following carcinogenic health risk index (C) is used to calculate the potential carcinogenic health risk of groundwater contamination:(6)C = CDI × SF

where SF is the slope coefficient. RfD and SF values are shown in Table 1 [46,47]. A high potential for carcinogenic health risk is indicated by an index value over the maximum allowed level of the ICRP-recommended carcinogenic health risk index of 5 × 10^−5^.

### 2.7. Statistical Analysis

Principal component analysis (PCA) is a multivariate technique for examining relationships among several quantitative variables [48,49]. The PCA reduces dimensionality of a dataset while maintaining its structure. The first principal component is the linear combination of the original variables that accounts for the greatest possible variance. The linear combination of the initial variables with the highest variance and no correlation to the previously established elements makes up each subsequent main component [50]. There are two primary factors to consider when choosing which elements to keep in the PCA: the first components with Eigenvalues greater than 1 are determined according to the Eigenvalue 1 criterion; amount of variation described (at least 70% to 80% of your variance should be explained by the selected factors) [51].

## 3. Results and Discussion

### 3.1. HM Concentrations in Groundwater Samples

Groundwater usually contains significant quantities of heavy metals. Heavy metal pollution in groundwater is mostly caused by the mineral oil industry, crop residues, urban and industrial pollutants, effluent, and fertilizer [52]. Abusing heavy metals can have disastrous effects, including intestinal inflammation and the development of several illnesses [53].

This study explains the concentration of twenty-three trace elements in groundwater samples collected from the Campania plain during the year 2015. Moreover, these data were used to evaluate the distribution, source apportionment, and health risk assessment to realize a clear groundwater hydrochemistry prospect in the study area.

The order of abundance values of trace element concentrations in overall groundwater samples of the study area were as follows: B > Fe > Al > Mn > Zn > Ba > Ni > As > Cu > V > Se > Pb > Cd, whereas the concentrations of Ag, Co, Cr, Ti, Mo, and Sn were below the detection limit (0.5 μg/L^−1^) (Table 2).

Among all the trace elements, the highest average concentration was obtained for boron (B). This metal has a high concentration in some sampled points (mean value 67.11 μg/L^−1^); the higher value of 1338 μg/L^−1^ was found in the study area of Caserta. In this case, the sample exceeded the limit value of D. Lgs. 152/2006, indicating that the groundwater is polluted by boron. This metal, an essential element in the human body, can damage the nervous and reproductive systems at high concentrations [54,55].

A high concentration was also obtained for Fe (29.79 µg/L^−1^). Although Fe is an important element in the human diet, it can cause damage if present in high concentrations [32]. In this study, the amount of Fe was indicated to be the highest in many sampled sites, but the results analyzed were above the permissible limit of 200 μg/L^−1^ given by Legislative Decree 152/06. These data can be attributed to the consistency of the geological crust. In reality, this metal is well known in biological systems and is widely distributed as a transition element in the crust of the earth. Under a range of natural environmental conditions, such as varying oxygen levels and aqueous pH levels, iron is easily able to modify its oxidation state. The greater amounts might be caused by oxidized, leached-in Fe-rich rocks in the groundwater system. Higher concentrations of Fe in CP groundwater may be caused by the presence of Fe-bearing minerals in the aquifer and their dissolution in the water. Additionally, the increased solubility of iron-bearing minerals brought on by the excessive use of dissolved oxygen by organic matter may pollute CP’s groundwater [56,57].

Aluminum (Al) was found to be 12.38 µg/L^−1^ (average concentration) in groundwater samples, with values lower than the permissible limit of 200 μg/L^−1^. In the municipalities of Giugliano, Scafati, Marigliano, and Sant’Agata dei Goti, the aluminum concentrations (198.36, 191.74, 118.03, and 111.18 μg/L^−1^, respectively) were found to be close to the permissible limit; whereas in the Avellino city, in the municipality of Bonito, the aluminum concentration (1193.00 μg/L^−1^) exceeded the permissible limit given by Legislative Decree 152/06. In high concentrations, metals such as aluminum can cause serious damage to cellular processes and behave as endocrine disruptors. Aluminum can interfere with the synthesis and metabolism of endogenous hormones, cellular function, and intercellular interactions [32,58].

Manganese (Mn), another heavy metal present in abundance in the geological crust, showed an average concentration of 13.9 µg/L^−1^. Mn in 2% of samples was found to exceed the permissible limit of 50 μg/L^−1^ Legislative Decree 152/06. High concentration levels of Mn were possibly due to municipal solid disposal for recycling [59].

The large number of CP groundwater samples were observed to be contaminated with Zn. This metal is beneficial to humans and other ecological entities; however, at higher concentrations, it can become toxic to humans’ biological systems. In this study, the average concentration was 12.8 µg/L^−1^; at some sampling sites, the concentrations were higher probably due to the use of urban areas as landfills and the application of fertilizers to agricultural fields. Because of discharges from exhaust pipes, using zinc as an additive to fossil fuels in car engines also contaminates the environment. Nickel (Ni), a potentially toxic element, was also found to exceed the permissible limits of 20 µg/L^−1^ in two sites: in the municipalities of Riardo, province of Caserta, where the amount of Ni was 263.8 µg/L^−1^, and in Castelpoto, province of Benevento, where the amount of Ni was 33.9 µg/L^−1^. In sites with higher groundwater, Ni concentrations from agricultural practices are highly developed. The use of Ni-enriched insecticides/fertilizers might be the primary reason for increased Ni concentrations in groundwater samples [60]. Ni and its components have been identified by the International Agency for Research on Cancer (IARC) as potential human carcinogens [61]. In fact, water contaminated with Ni has been reported to cause numerous health-related issues such as conjunctivitis, asthma, colitis, kidney damage, and cancer [62].

Arsenic (As) has been classified as a human carcinogen that results in skin and lung cancer [63]. Significant neurological, cardiovascular, respiratory, and reproductive damage can occur from exposure to high concentrations [64,65]. The arsenic concentrations found in the groundwater were lower than the legal limit of 10 µg/L^−1^ given by Legislative Decree 152/06, except for Cancello Arnone, Mondragone, and Sessa Aurunca sites, where the concentrations were 145.3 µg/L^−1^; 102.5 µg/L^−1^, and 25.4 µg/L^−1^, respectively.

Other measured elemental species (Cu, V, Se, Pb, Cd, Ag, Be, Co, Cr, V, Ti, Mo, and Sn) were found in trace amounts and within the respective permissible limits given by Legislative Decree 152/06.

### 3.2. HM Distributions in Groundwater

Figure 2 shows the principal results of the distribution of heavy metals, whereas in the Appendix A, distribution data of all the other metals present in CP are reported. The Jenks optimization approach is used to classify the data in this study; this technique is based on the naturally occurring categories in the data and determines break points by choosing the class breaks that best group similar data and maximize the differences between classes. With boundaries placed where there are relatively large jumps in the data values, the characteristics are split into classes.

The spatial distribution of heavy metals is heterogeneous in CP. Boron concentration in groundwater samples shows a relatively higher percentage throughout the regional territory (Naples 35%, Salerno 37%, Caserta 38%, Benevento 37%, and Avellino 44%, respectively). This abundance might be due to the geogenic origin of boron and the organically bound form of boron. Aluminum data have shown that the distribution of this metal is widespread in all the provinces of Campania. This distribution may be dependent on the fact that Al is not easily impacted by weathering processes and is mostly related to forming processes and the presence of oxides in clastic materials. These data prove that Al remains the most stable element in ecological systems [66].

Fe also has a heterogeneous distribution in the northwest of the research area. Weathering of minerals and rocks containing Fe and Mn is the natural source of these elements in groundwater. Common human activities that cause the concentration to increase include metal processing, iron and steel processing, and coal mining [67].

In the region, concentrations of Pb and Ni were observed. In particular, Ni is more distributed in the territory of Naples, Caserta, and Benevento. It demonstrates the impact of byproducts of human activity, such industrial effluents, automobile emissions, e-waste, and natural factors [67]. Concentrations of As in groundwater samples are relatively abundant, which might be due to the geogenic origin and due to the presence of the natural background values of this metal in the Campania, in particular in the area of Naples (Phlegrean Fields) and Caserta. The aquifers, which are made up of volcanic and pyroclastic rocks, have natural properties that account for this concentration. The high arsenic level of the aquifers is due to their volcanic origin, particularly in the groundwater of the “Phlegrean Fields” [68].

### 3.3. HM Source Identification in Groundwater Samples

In this study, PCA was employed to support in the identification of the origin of pollutants. A 3-D plot of the PCA loadings is proposed in Figure 3, and the correlation among the eleven heavy metals is easily observed. Just as expected, four factors were obtained (Table 3). Between these, the highest factor with 18% is dominated by As and B (PCA1). This group could be monitored by natural (geogenic) factors, and the presence of As in the groundwater samples is presumably due to the reductive dissolution of As-rich FeO (OH) [69], whereas the B component is typically related to the organically bound, clay- and mineral-fixed fraction [70]. The second factor (PCA2) was dominated by Al, Ni, Zn, Ba, and Cu and is accounts for 14% of the total variance. This component was influenced by both geogenic and anthropogenic activities. In fact, the presence of heavy metals such as Al and Ni indicated anthropogenic pollution from domestic and agricultural sources. More specifically, Ni is linked to release from stainless steel and alloy product industries. Other metals such as Zn, Ba, and Cu may be due to natural geological origin. In fact, these elements are the immobile metals because they are usually adsorbed by clay minerals as well as the organic materials in the soil [70,71]. PCA3 was best represented by Fe and Mn, which accounted for 11% of the total variance and which implies the natural sources of these elements. In fact, Mn usually occurs naturally as a mineral from sedimentary rocks, and bacterial activities on Fe and Mn are also responsible for the release of these elements [72]. The PCA4 was instead dominated by V and Se, and these components seemed to be associated with the earth’s crust and geological formations [73]. These components accounted for 10% of total variance.

### 3.4. HEI and HPI Results

Estimation of groundwater pollution has been carried out by applying the typically used contamination indices. The HPI values for all provinces of the Campania plain ranged from 0.002 to 9.1 for Avellino, 0.003 to 10.7 for Caserta, 0.002 to 10.1 for Naples, 0.002 to 14.3 for Benevento, and 0.002 to 11.1 for Salerno, respectively. The HEI values ranged from 0.0008 to 0.454 for Avellino, 0.0008 to 0.428 for Caserta, 0.0008 to 0.433 for Naples, 0.0008 to 0.370 for Benevento, and 0.0008 to 0.422 for Salerno, respectively.

Based on the results, the HPI and HEI values determined for the groundwater samples collected in the Campania plain were less than 100 and did not exceed the permissible value. Lower values demonstrate that byproducts of human activity, such as urban discharge and fertilizer waste, pose no harm to the groundwater system in terms of heavy metal contamination. In the same way, heavy metal evaluation index (HEI) values were also determined to define the heavy metal contamination load in the groundwater. All stations in the present study belonged to the low category, indicating that there is a low level of heavy metals and, therefore, a low degree of pollution in the study area.

### 3.5. Human Health Risk Assessment

#### 3.5.1. Non-Carcinogenic Risk Assessment

In many places (rural and urban), groundwater contamination is a significant environmental and health concern, and heavy metals are one of the main causes [33].

Therefore, it is necessary to evaluate the human health hazard because of possible heavy metal contamination associated with the use of water from these wells. Health risk assessment is now one of the best procedures to examine the potential risk of exposure to HM for humans and provides essential public health information.

Ingestion of food, inhalation of aerosols, and ingestion of drinking water are just a few of the ways that heavy metals can be ingested and pose a health risk. This study showed the non-carcinogenic risk (HQ) estimates for As, Al, Cd, Cr, Cu, Ni, Pb, Zn, Fe, Mn, and Ba. The HQ for most heavy metals studied has potential to pose non-carcinogenic health impacts on living beings. It was > 1 HQ ≤ 5, indicating a low risk. The HQ values for heavy metals were less than 1 in the adult age group. For the age class of children, the HQ of As metal was more than 1 (As 1.09). Infants’ HQ values in Metals rise to 1.63 as well. Heavy metals may be related to a possible non-carcinogenic danger if the HQ is higher than 1 [73]. As a result, children and infants in the Campania plain area may develop non-cancerous disorders as a result of the high values of HQ found in As. The mean HQ index values for the adult group were 0.311, 0.002, 0.120, 0.021, 0.002, 0.004, 0.022, 0.001, 0.002, 0.002, and 0.002, respectively. For the children group, the mean HQ index values were 1.09, 0.009, 0.420, 0.076, 0.010, 0.014, 0.079, 0.004, 0.009, 0.008, and 0.007, respectively. For the infant group, the mean HQ index values were 1.63, 0.013, 0.630, 0.115, 0.015, 0.021, 0.119, 0.006, 0.014, 0.013, and 0.011, respectively. Therefore, the order of toxicity of HM mean concentrations for groundwater were found in the order of As > Cd > Pb > Cr > Ni > Cu > Fe > Mn > Ba > Zn. Moreover, the hazard index (HI) is the cumulative sum of all examined metals and was used to compute overall non-carcinogenic risk. The average value for adults, children, and infants was found to be 0.19, 0.66, and 0.99, respectively.

For all three groups, the values of the HI ratio < 1, showing that the cumulative non-carcinogenic risk is relatively low, even if the groups of children and infants have higher values than that of adults. This result signifies that children and infants were more vulnerable to health risks from heavy metal contamination.

#### 3.5.2. Carcinogenic Risk Assessment

Based on the supplied slope factor, incremental lifetime cancer risk (ILCR) estimates have been determined for metal(loid)s As, Ni, and Pb. USEPA recommendations for both age groups of the population state that when ILCR < 1.0 × 10^−6^, there is no risk; if the ILCR values are between 1.0 × 10^−6^ and 1.0 × 10^−4^, the carcinogenic risk is acceptable; and if ILCR values > 1.0 × 10^−4^, the risk is not acceptable [74,75].

Human cancer risk can be determined by heavy metals including As, Ni, and Pb; prolonged exposure can raise the probability of developing cancer [33,35]. Moreover, carcinogenicity risk was estimated for these heavy metals. In fact, if several carcinogenic metals are found in the system, the toxicity risk from all carcinogens is added to evaluate the combined carcinogenic impact. The specific mean values for As, Ni, and Pb for adults range from 1.40 × 10^−4^, 1.22 × 10^−5^, and 1.34 × 10^−6^, respectively; whereas for children, the values range from 1.02 × 10^−4^, 1.31 × 10^−4^, and 1.71 × 10^−6^, respectively. For infants, the specific mean values range from 1.3 × 10^−4^, 1.5 × 10^−4^, and 1.04 × 10^−4^, respectively. The results estimated for the groundwater of the CP have shown that all three metals analyzed do not exceed the guidelines limits; therefore, the CR risk can be considered acceptable in this study area.

As a result, the observed heavy metal concentration is below the standard reference ranges [74]. The groundwater was indicated as safe for drinking or domestic purposes, as the samples also demonstrate some indices such as hazard index (HI) or cancer risk (CR) that have been calculated and analyzed.

## 4. Conclusions

This study is part of a large research project which involves the sampling and analysis of groundwater sampled in the following aquifers: coastal plains (GAR, VCP, VES, SAR and SEL), volcanic districts (PHLE and VES), and carbonate massifs (MAS and LAT) of the Campania Plain for the determination of the most emerging environmental pollutants. In this study, we precisely collected 1093 groundwater samples and assessed the groundwater quality for 23 heavy metals using a variety of evaluation indices to determine the potential health concerns for humans. This study confirmed that the order of the abundance values of trace element concentrations in the overall groundwater samples of the study area were as follows: B > Fe > Al > Mn > Zn > Ba > Ni > As > Cu > V > Se > Pb > Cd, whereas the concentrations of Ag, Co, Cr, Ti, Mo, and Sn were below the detection limit (0.5 μg/L^−1^) given by Legislative Decree 152/06. Total concentrations of heavy metals have shown a somewhat heterogeneous distribution of these compounds in the groundwater samples of the Campania Plain, and PCA analysis was used to find the most likely source of the heavy metals. The PCA 1, 3, and 4 were attributed mainly to natural sources, whereas the PCA 2 was attributed to anthropogenic and geochemical activities. Examination of heavy metals indices (HPI and HEI) demonstrated that in all sampled sites, HPI < 100 does not exceed the permissible value. Similarly, HEI values showed the heavy metal contamination load in the groundwater. In all stations, the HEI value belonged to the low category, indicating that the study area is characterized by a low-level heavy metal and, therefore, a low degree of pollution. The HQ values for heavy metals were less than 1 in the adult age group, whereas the HQ value was found to be higher for As in children and infants, which implies adverse health impacts which are non-carcinogenic. CR results showed that all three metals analyzed did not exceed the guidelines limits. Therefore, the CR risk can be considered to be acceptable in this study area. These research data may furnish insights into the groundwater quality of the Campania plain and help to formulate protective measures to reduce metal contamination in the water system.

## Figures and Tables

**Figure 1 ijerph-20-01693-f001:**
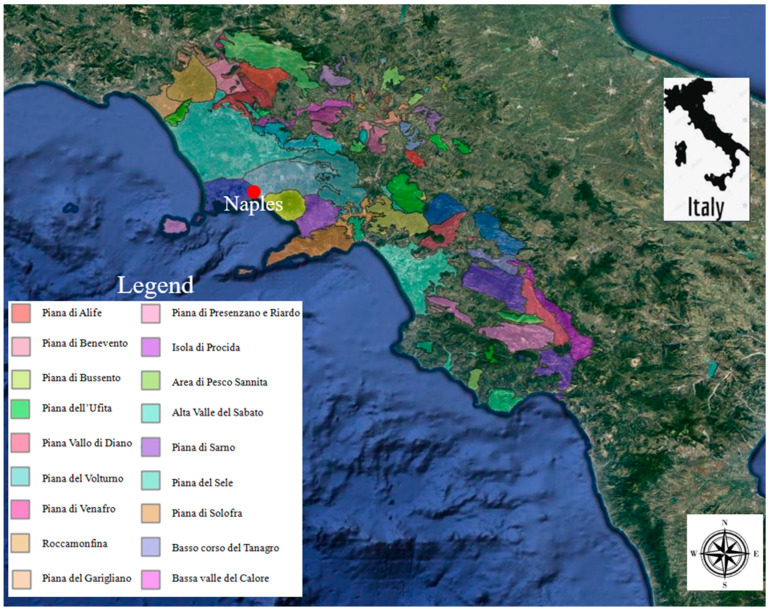
Map of the study area of Campania Region.

**Figure 2 ijerph-20-01693-f002:**
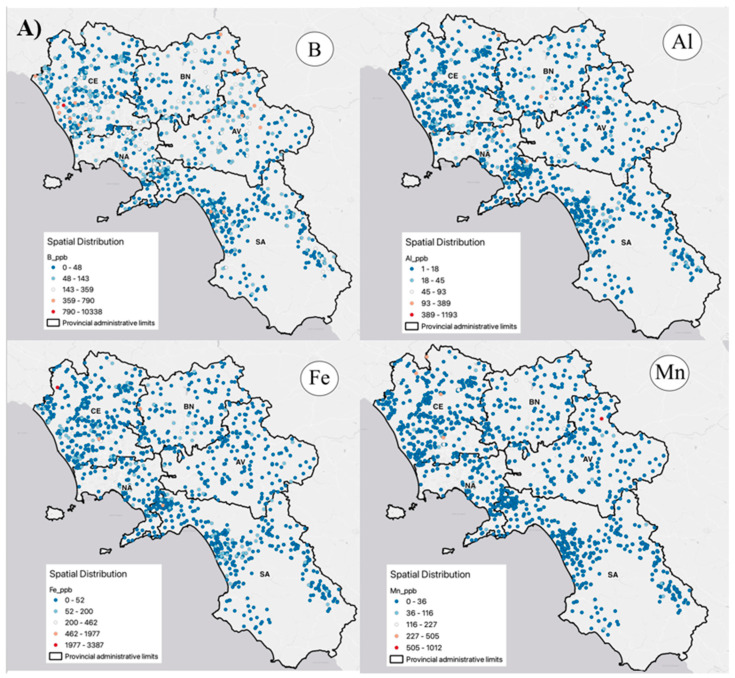
Spatial distribution of heavy metals in groundwater of Campania Plain. In (**A**) heavy metals B, Al, Fe and Mn are represented, while in (**B**) heavy metals Pb, Ni and As are represented.

**Figure 3 ijerph-20-01693-f003:**
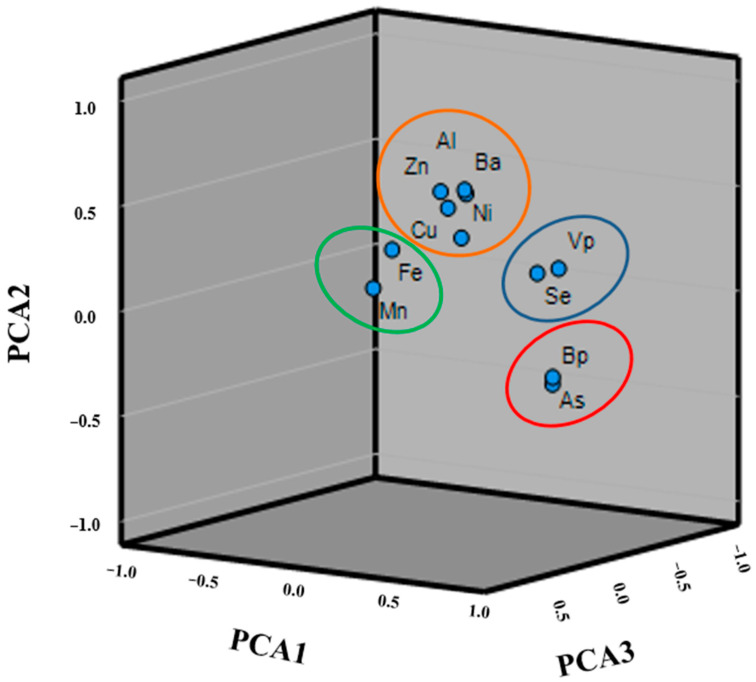
Principal component analysis (3-D space plot) for heavy metal distribution.

**Table 1 ijerph-20-01693-t001:** Standard values of RfD and SF in the study [46,47]. Table 1 is recreated by the authors.

Noncarcinogen	RfD (mg/(kg × Day))	Carcinogen	SF (kg × Day/mg)
Al	0.14	As	1.5
Fe	0.3	Cd	0.63
Zn	0.3	Cr	42
Cu	0.04	Ni	0.84
Mo	0.005	Pb	0.042
Ni	0.02		
Mn	0.14		
Co	0.0003		
Ba	0.2		
Pb	0.0014		
Cd	0.0005		
Cr	0.003		
As	0.0003		

**Table 2 ijerph-20-01693-t002:** Minimum, Maximum, and Mean concentrations of heavy metals in groundwater samples of Campania Plain.

Heavy Metals (μg/L^−1^)	Minimum	Maximum	Mean	D. Lgs. 152/2006 Limits
B	0.57	10,338.0	67.1	1000
Fe	0.75	3387.3	29.7	200
Al	0.51	1193.0	12.3	200
Mn	0.82	1012.2	13.9	50
Zn	0.85	786.2	12.8	3000
Ba	0.97	518.4	14.1	n.a.
Ni	0.61	263.8	3.66	20
As	0.78	145.3	3.56	10
Cu	0.83	85.9	3.87	1000
V	0.57	81.9	5.26	n.a.
Se	0.59	12.9	1.58	10
Pb	0.90	7.5	1.22	10
Cd	0.53	6.2	2.14	5

n.a.—not applicable; D. Lgs. 152/2006 Limits: Limits Legislative decree 152/2006.

**Table 3 ijerph-20-01693-t003:** Results of principal component analysis.

Variables	Components			
	PC1	PC2	PC3	PC4
Al	0.186	**0.521**	−0.052	0.065
As	**0.874**	−0.249	0.174	−0.172
Fe	0.152	0.339	**0.548**	0.325
Ni	0.110	**0.295**	−0.118	−0.220
Cu	0.074	**0.443**	−0.053	−0.445
Se	0.279	0.082	−0.540	**0.451**
Mn	0.050	0.148	**0.566**	0.433
Zn	0.080	**0.532**	0.021	−0.379
B	**0.855**	−0.288	0.149	−0.185
Ba	0.209	**0.549**	−0.005	0.138
V	0.363	0.134	−0.465	**0.443**

PC: Principal component. Bold indicates the most abundant component.

## Data Availability

The datasets obtained and analyzed in the current study are available from the corresponding author on reasonable request.

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
