# Peer review of "Heavy Metals in Groundwater of Southern Italy: Occurrence and Potential Adverse Effects on the Environment and Human Health"

_ijerph, 2023, doi:10.3390/ijerph20031693_

Round 1
Reviewer 1 Report
The authors have undertaken extensive work on the contamination caused by heavy metals in groundwater of Campania Plain in Southern Italy, their occurrence and potential adverse effects on the environment and human health.
The results are interesting, however, there are still some minor problems to be addressed.
Some issues listed below have to be solved:
1. Naples or Napoli, please be the same in the affiliations and the entire text.
2. pp. 2, line 75 avoid redundant repetitions…. Campania Plain, Southern Italy; lowercase or uppercase letter in Southern. Also check for typographical errors throughout the text.
3. pp. 4, 6, 7, 8… please use the same unit: μg/L, μg L−1, μg /L−1 (?)
4. pp. 5 line 169 Heavy metal pollution (HPI) change to Heavy metal pollution index (HPI)
5. Figures 2, 3, as well as in the supplementary file, should be improved in resolution and readability.
6. The English language, grammar and sentence structures need to be improved.
7. The authors might find interesting to cite the newer article regarding the heavy metal pollution index (HPI):
J. Milivojević, D. Krstić, B. Šmit, V. Djekić, Bull. Environ. Contam. Toxicol. (2016) 97:737–742
doi: 10.1007/s00128-016-1918-0
Reviewer 2 Report
General comments
Research that fits the scope of IJERPH. Some detail is missing, and all the comments need to be addressed before publication
Specific comments
Lines 11-22. Add information on the sampled aquifer-systems in the abstract, and conclusions
Line 19. Specify which groups. Remind the information to the reader
Lines 47-49. “The chemical-physical characteristics of the aquifer, precipitation 47 periodicity, the quality of infiltrating water, and the residence time are the typical aspects 48 that regulation the presence of HM in groundwater”. Add recent review papers that deals with surface/groundwater interaction, water quality, and resident time
- Medici, G. and Langman, J.B., 2022. Pathways and Estimate of Aquifer Recharge in a Flood Basalt Terrain; A Review from the South Fork Palouse River Basin (Columbia River Plateau, USA). Sustainability, 14(18), p.11349.
- Ahmed, I., Tariq, N. and Al Muhery, A., 2019. Hydrochemical characterization of groundwater to align with sustainable development goals in the Emirate of Dubai, UAE. Environmental Earth Sciences, 78(1), pp.1-17.
Lines 76-77. “To the best of our knowledge, no previous studies report the contamination of GW by HMs in Southern Italy and in the Campania Plain”. I agree with this statement for the specific contaminants that you treat. But, you should specify that the Campania region is the most studied in Italy for contamination risk mapping. Such studies of geostatistics were produced by your university, see below one example:
- Ducci, D., 1999. GIS techniques for mapping groundwater contamination risk. Natural Hazards, 20(2), pp.279-294.
Lines 87. “Material ad methods”. Good introduction for concepts, sampling procedures and equations. Please, insert more detail on the mapping techniques that you have used
Lines 89-122. Add more detail on the hydrogeological setting. What about types of rocks and aquifer units?
Line 122. Add more detail on the aquifer systems that you have sampled. All samples from one system? Depth of investigation?
Line 262. “Consistency of the geological crustal”. Sentence unclear from a geological point of view. Please, revise
Lines 434-457. Add information on the sampled aquifer-system in the abstract, and conclusions
Line 474. Add the relevant papers suggested above to complete your introduction
Figures and tables
Figure 1. Insert at least the position of the City of Naples
